# Flexible 5G New Radio LDPC Encoder Optimized for High Hardware Usage Efficiency

**Vladimir L. Petrović** [1,*][iD]**, Dragomir M. El Mezeni** [1] **and Andreja Radošević** [2]

[1]   School of Electrical Engineering, University of Belgrade, Bulevar kralja Aleksandra 73, 11120 Belgrade, Serbia; elmezeni@etf.bg.ac.rs

[2]   Tannera LLC, 401 Wilshire Blvd, 12th Floor, Santa Monica, CA 90401, USA; andreja@tannera.io

*   Correspondence: petrovicv@etf.bg.ac.rs

**Abstract:** Quasi-cyclic low-density parity-check (QC–LDPC) codes are introduced as a physical channel coding solution for data channels in 5G new radio (5G NR). Depending on the use case scenario, this standard proposes the usage of a wide variety of codes, which imposes the need for high encoder flexibility. LDPC codes from 5G NR have a convenient structure and can be efficiently encoded using forward substitution and without computationally intensive multiplications with dense matrices. However, the state-of-the-art solutions for encoder hardware implementation can be inefficient since many hardware processing units stay idle during the encoding process. This paper proposes a novel partially parallel architecture that can provide high hardware usage efficiency (HUE) while achieving encoder flexibility and support for all 5G NR codes. The proposed architecture includes a flexible circular shifting network, which is capable of shifting a single large bit vector or multiple smaller bit vectors depending on the code. The encoder architecture was built around the shifter in a way that multiple parity check matrix elements can be processed in parallel for short codes, thus providing almost the same level of parallelism as for long codes. The processing schedule was optimized for minimal encoding time using the genetic algorithm. The optimized encoder provided high throughputs, low latency, and up-to-date the best HUE.

**Keywords:** channel coding; 5G new radio; low-density parity-check (LDPC) codes; encoder architecture; hardware usage efficiency; circular shifter; genetic algorithm optimization





## 1. Introduction

Fifth-generation wireless technology standard for broadband cellular networks (5G NR) [1] introduces LDPC codes [2] for data channel coding. Due to their excellent error-correcting performance and the possibility of achieving high parallelization in decoding procedures, LDPC codes outperform turbo codes [3], as their predecessor in 3G and 4G LTE [4,5]. The main channel coding challenges posed by novel 5G-centric applications are multigigabit throughputs, low latency (in certain use cases), and high flexibility while keeping high error-correcting performance. In addition to that, the channel coding solution should support incremental redundancy hybrid automatic repeat request (IR–HARQ) procedures. LDPC codes can face all those challenges better than turbo codes. Most importantly, they have higher coding gain and have better error-correcting performance in the error floor region [5].

LDPC codes are linear block codes, meaning that they are completely defined by their parity check matrices (PCM). One of the reasons for their increasing adoption in many communication standards is the computational efficiency of the decoding procedure, which is achieved by designing the sparse PCM. In addition, the PCM of practical LDPC codes is always structured in a way that can achieve high parallelization of both the encoder and the decoder. On the other hand, turbo code decoding is inherently serial. Although it can be parallelized to some extent, high degrees of parallelism are hardly achievable. Moreover, the throughput of turbo decoders is usually not dependent on the code rate, whereas

LDPC decoder throughput increases along with the code rate since PCM for higher code rate is usually smaller than the PCM for low code rate. This property enables achieving higher peak throughput at high spectral efficiencies and high values of signal-to-noise ratio (SNR) [5].

The most common structured LDPC family is known as quasi-cyclic (QC) LDPC codes [6–8]. Their PCM consists of circularly shifted identity submatrices or square zero submatrices. Such submatrices are frequently called circulant permutation matrices (CPM) [8,9]. The size of CPMs is referred to as lifting size or lifting factor (Z) [5]. QC–LDPC codes are usually designed based on the so-called base graph (BG) matrix, which defines a macroscopic structure of a code. The final PCM is constructed by placing CPMs at appropriate positions defined by the BG matrix. Since CPMs are either circularly shifted identity matrices or zero matrices, the PCM can be stored as a matrix of integer numbers that represent corresponding shift values or zero matrix identification. Such description is frequently called a permutation matrix [7] or exponent matrix [9] and is very much convenient for 5G NR since the number of supported codes is very high. The great diversity of code rates and codeword lengths additionally requires a unification of code descriptions; therefore, the representation using exponent matrices is essential [5,9].

Another important property of an LDPC code is its regularity. An LDPC code is regular if all columns in the PCM have the same number of nonzero entries, i.e., if they all have the same weights, and if all rows in the PCM have the same weights. However, the error-correcting performance of such LDPC codes is usually not as good as the performance of irregular LDPC codes [10]. Hardware that supports irregular LDPC codes is usually more challenging to design since irregular code structure can significantly affect HUE [11]. Nevertheless, 5G NR introduces a large number of irregular LDPC codes with various codeword lengths and code rates. Therefore, hardware that supports 5G NR codes must provide a high level of flexibility [11].

Multiple encoding methods for LDPC codes have been developed [12–35]. A straightforward method is to use a generator matrix for encoding [12–16]. Such a method is computationally very expensive since, in general, the generator matrix is not sparse. Reduction of complexity can be obtained for systematic codes if the PCM is divided into two parts [17–19]. This way, the encoding can be performed by multiplication with two smaller matrices of which the first one is sparse, but the second one is dense. An approximately linear complexity can be obtained by using the Richardson and Urbanke (RU) method [20]. This method is based on the offline preprocessing algorithm that permutes columns and rows of the PCM so that the permuted matrix has a specific structure that is convenient for encoding. If the proper restructuring is achieved, the encoding can be performed mainly by multiple sequential forward substitutions and one multiplication with a small matrix, which is why this is a widely used approach [21–27]. A hybrid method has been used in [28,29], which proposes the calculation of one part of the parity bits using the generator matrix and another part using the RU method. However, many practical codes can be encoded using only forward substitutions, due to the convenient code structure [30–35]. Of note, 5G NR codes belong to this class of codes.

This paper focuses on the hardware implementation of a flexible 5G NR encoder that supports all codes defined by the standard and achieves high throughput for a wide spectrum of codes while keeping hardware resources as low as possible. The encoding process is optimized for high hardware usage efficiency expressed in terms of achieved throughput divided by used hardware resources. Optimizing HUE can minimize required hardware resources for an arbitrary target throughput. As long as the latency of the system is acceptable, it is always more efficient to employ multiple optimized encoders in parallel and achieve the required throughput than to use a highly parallel implementation that takes significantly more resources. In general, the encoding latency is usually not an issue since the soft-decision LDPC decoding is usually a bottleneck of the entire physical layer [28]. In addition to that, resource-efficient hardware design is strongly linked to energy efficiency. Idle processing blocks consume leakage energy, which is an increasingly

significant energy component in modern technology processes [36]. Therefore, optimizing HUE can lead to a more energy-efficient design.

In general, 5G NR LDPC codes are highly irregular and their base graph matrices are mostly sparse. This property drastically reduces the efficiency of highly parallel architectures since many hardware processing units frequently remain idle. Therefore, in this work, a partially parallel architecture is proposed in order to obtain high HUE.

The main contributions of this paper are (1) high flexibility of the LDPC encoder that supports all codes from 5G NR. This is primarily obtained by the proper design of a circular shifting network, which is the key hardware processing unit in the system; (2) by exploiting partially parallel processing, the circular shifter and other hardware processing units are rarely idle, which highly increases the hardware usage efficiency; and (3) the encoder is capable of changing the encoding schedule in runtime so as to exploit high parallelism both for codes with high values of lifting size and for codes with smaller lifting sizes. Encoding schedules are optimized for minimum encoding time, which provides a minimum latency and the maximum throughput for the observed architecture. Such optimization significantly increases the HUE.

The rest of the paper is organized as follows. Section 2 presents an overview of commonly used methods for LDPC encoding. In Section 3, 5G NR LDPC encoding is discussed, first by introducing the structure of 5G NR codes and then by describing the efficient encoding algorithm. Section 4 introduces a critical comparison of various approaches for the implementation of the aforementioned algorithm. Afterward, an optimization method for the selected implementation strategy is presented. The proposed encoder architecture is described in Section 5, whereas results are shown in Section 6. Section 7 concludes the paper.

## 2. Encoding of LDPC Codes

Depending on the code structure and properties, various methods have been developed for encoding LDPC codes [12–35]. This section briefly introduces mostly used methods that have been applied in LDPC encoders.

### 2.1. Straightforward Encoding With Generator Matrix Multiplication

As previously mentioned, LDPC codes are linear block codes. They are completely defined by their parity check matrices, which are designed to be sparse so as to obtain smaller decoding complexity. The parity check matrix $\mathbf{H}$ and the codeword vector $\mathbf{x} = [x_1 x_2 \ldots x_n]$ must satisfy the following equation:

$$\mathbf{x}\mathbf{H}^{\mathrm{T}} = 0. \tag{1}$$

Most practical LDPC codes are systematic, meaning that their codeword can be represented as follows:

$$\mathbf{x} = \begin{bmatrix} \mathbf{i} & \mathbf{p} \end{bmatrix}, \tag{2}$$

where the vector $\mathbf{i} = [i_1 i_2 \ldots i_k]$ represents a sequence of information bits and the vector $\mathbf{p} = [p_1 p_2 \ldots p_m]$ represents a sequence of parity bits. The encoding process implies mapping the vector $\mathbf{i}$ to the vector $\mathbf{x}$. However, in general, the code does not have to be systematic, in which case information and parity bits are not divided into separate groups. In any case, the straightforward encoding method is to multiply the information bits vector with the generator matrix $\mathbf{G}$ as

$$\mathbf{x} = \mathbf{i}\mathbf{G}. \tag{3}$$

The generator matrix can be obtained by solving Equation (4) using the Gaussian elimination.

$$\mathbf{G}\mathbf{H}^{\mathrm{T}} = 0 \tag{4}$$

The obtained generator matrix is generally dense, which causes quadratic encoding complexity ($O(n^2)$) [20]. However, when the generator matrix is precalculated, the required

matrix multiplication for specific codes can be efficiently achieved in hardware. Therefore, several previous encoders have been based on this method [12–16].

### 2.2. Partitioned PCM Two-Step Encoding

In cases when the LDPC code is systematic, the PCM can be partitioned into two submatrices ($\mathbf{H} = \begin{bmatrix} \mathbf{H}_1 & \mathbf{H}_2 \end{bmatrix}$), where the first submatrix is of size $m \times k$, and the second submatrix is of size $m \times m$. By transposing both the left-hand side and the right-hand side of the equation, Equation (1) becomes

$$\mathbf{H}\mathbf{x}^{\mathrm{T}} = 0, \tag{5}$$

which can further be written as

$$\begin{bmatrix} \mathbf{H}_1 & \mathbf{H}_2 \end{bmatrix} \begin{bmatrix} \mathbf{i} & \mathbf{p} \end{bmatrix}^{\mathrm{T}} = \mathbf{H}_1 \mathbf{i}^{\mathrm{T}} + \mathbf{H}_2 \mathbf{p}^{\mathrm{T}} = 0. \tag{6}$$

Parity bits can now be calculated as follows:

$$\mathbf{p}^{\mathrm{T}} = \mathbf{H}_2^{-1} \mathbf{H}_1 \mathbf{i}^{\mathrm{T}} = 0. \tag{7}$$

This way, the calculation is simplified since the submatrix $\mathbf{H}_1$ is sparse, and the vector $\mathbf{H}_1 \mathbf{i}^{\mathrm{T}}$ can be calculated with linear complexity ($O(n)$). However, the matrix $\mathbf{H}_2^{-1}$ is dense and therefore leads to significantly high hardware cost in terms of computational complexity and memory requirements. Nevertheless, when properly exploiting the matrix structure, it is possible to achieve significant hardware cost reduction for certain codes [17–19].

### 2.3. Richardson–Urbanke LDPC Encoding Method

Richardson and Urbanke [20] proposed an LDPC encoding method that provides an approximately linear computational complexity. The proposed method consists of two steps: (1) offline preprocessing and (2) encoding. During the preprocessing step, the PCM is transformed to the approximate lower triangular (ALT) form in such a way that codewords can be represented in a systematic form as follows:

$$\mathbf{x} = \begin{bmatrix} \mathbf{i} & \mathbf{p}_1 & \mathbf{p}_2 \end{bmatrix}, \tag{8}$$

where $\mathbf{p}_1$ represents a vector of $g$ parity bits, whereas $\mathbf{p}_2$ represents a vector of remaining $(m - g)$ parity bits.

After the preprocessing step, the PCM has the following form:

$$\mathbf{H} = \begin{bmatrix} \mathbf{A} & \mathbf{B} & \mathbf{T} \\ \mathbf{C} & \mathbf{D} & \mathbf{E} \end{bmatrix}, \tag{9}$$

where submatrices $\mathbf{A}$ (of size $(m - g) \times (n - m)$), $\mathbf{B}$ (of size $(m - g) \times g$), $\mathbf{C}$ (of size $g \times (n - m)$), $\mathbf{D}$ (of size $g \times g$), and $\mathbf{E}$ (of size $g \times (m - g)$) are sparse matrices, whereas the submatrix $\mathbf{T}$ (of size $(m - g) \times (m - g)$) is a sparse lower triangular matrix. The preprocessing transformation is performed by permuting rows and columns inside the PCM. Appropriate permutations can be obtained by greedy algorithms described in [20], which provide $g$ to be as small as possible. In addition, it is required that the matrix $-\mathbf{E}\mathbf{T}^{-1}\mathbf{B} + \mathbf{D}$ stays nonsingular in the Galois field GF(2).

If the PCM transformation is successfully completed, parity bits can be calculated using the following equations:

$$\mathbf{p}_1^{\mathrm{T}} = \left( -\mathbf{E}\mathbf{T}^{-1}\mathbf{B} + \mathbf{D} \right)^{-1} \left( -\mathbf{E}\mathbf{T}^{-1}\mathbf{A} + \mathbf{C} \right) \mathbf{i}^{\mathrm{T}}, \tag{10}$$

$$\mathbf{p}_2^{\mathrm{T}} = -\mathbf{T}^{-1} \left( \mathbf{A}\mathbf{i}^{\mathrm{T}} + \mathbf{B}\mathbf{p}_1^{\mathrm{T}} \right). \tag{11}$$

Multiplications with **A**, **B**, **C**, and **E** are multiplications with sparse matrices, whose computational complexity is linear ($O(n)$). Additions also have linear complexity. Since **T** is a lower triangular matrix, all multiplications with $\mathbf{T}^{-1}$ can be replaced with a forward substitution, given that $\mathbf{T}^{-1}\mathbf{a} = \mathbf{b}$ is equivalent to the system $\mathbf{a} = \mathbf{Tb}$. Since **T** is also sparse, the performed forward substitution is a process of linear complexity. The only quadratic complexity calculation is multiplication by dense matrix $\left(-\mathbf{ET}^{-1}\mathbf{B} + \mathbf{D}\right)^{-1}$ ($O(g^2)$), which is why the transformation algorithm searches for permutations that will provide small $g$, preferably $g \sim \sqrt{n}$. Such approximately linear encoding complexity is the main reason why many LDPC encoders are essentially based on the RU method [20–27]. The reported weaknesses of this method are the necessity for multiple consequent operations, which generally leads to long critical paths in hardware implementations [28], and low potential for encoding flexibility [14].

### 2.4. Straightforward and RU Hybrid Encoding

In order to avoid multiplication with a dense matrix $\left(-\mathbf{ET}^{-1}\mathbf{B} + \mathbf{D}\right)^{-1}$ from (10), Cohen and Parhi [28] proposed a hybrid encoding of IEEE 802.3an codes, which instead uses generator matrix for calculation of first $g$ parity bits ($\mathbf{p}_1$) and then uses RU's relation from (11) to calculate remaining parity bits ($\mathbf{p}_2$). Such an approach allows easier transformation of the PCM since the constraint for nonsingularity of the matrix $-\mathbf{ET}^{-1}\mathbf{B} + \mathbf{D}$ is removed. Additionally, the critical path can be shortened, which can lead to faster hardware implementation. The hybrid encoding has also been used for LDPC codes in space applications [29].

### 2.5. Forward Substitution-Based Encoding

Many standardized LDPC codes have a structure that can facilitate even more efficient encoding than any of the previously described methods. For example, codes from Wi-Fi 802.11n/ac/ax have a double diagonal structure of the right-hand side of the PCM, whereas the left-hand side has a nonspecific conventional quasi-cyclic structure [37], as shown in (12).

$$\mathbf{H} = \begin{bmatrix} \mathbf{h}_{1,1} & \mathbf{h}_{1,2} & \cdots & \mathbf{h}_{1,k_b} & \mathbf{I}^{(1)} & \mathbf{I} & 0 & \cdots & 0 & 0 & \cdots & 0 \\ \mathbf{h}_{2,1} & \mathbf{h}_{2,2} & \cdots & \mathbf{h}_{2,k_b} & 0 & \mathbf{I} & \mathbf{I} & \cdots & 0 & 0 & \cdots & 0 \\ \vdots & \vdots & \ddots & \vdots & \vdots & \vdots & \vdots & \ddots & \vdots & \vdots & \ddots & \vdots \\ \mathbf{h}_{r-1,1} & \mathbf{h}_{r-1,2} & \cdots & \mathbf{h}_{r-1,k_b} & 0 & 0 & 0 & \cdots & \mathbf{I} & 0 & \cdots & 0 \\ \mathbf{h}_{r,1} & \mathbf{h}_{r,2} & \cdots & \mathbf{h}_{r,k_b} & \mathbf{I} & 0 & 0 & \cdots & \mathbf{I} & \mathbf{I} & \cdots & 0 \\ \mathbf{h}_{r+1,1} & \mathbf{h}_{r+1,2} & \cdots & \mathbf{h}_{r+1,k_b} & 0 & 0 & 0 & \cdots & 0 & \mathbf{I} & \cdots & 0 \\ \vdots & \vdots & \ddots & \vdots & \vdots & \vdots & \vdots & \ddots & \vdots & \vdots & \ddots & \vdots \\ \mathbf{h}_{m_b,1} & \mathbf{h}_{m_b,2} & \cdots & \mathbf{h}_{m_b,k_b} & \mathbf{I}^{(1)} & 0 & 0 & \cdots & 0 & 0 & \cdots & \mathbf{I} \end{bmatrix} \tag{12}$$

In (12), $\mathbf{h}_{j,k}$ represents a single CPM in QC–LDPC code's PCM, which can be either circularly shifted identity matrix or zero matrix, $k_b$ is the number of information bit groups each of which has $Z$ bits, whereas the $m_b$ is the number of parity check equation groups each of which contains $Z$ parity check equations. **I** is an identity matrix. In general, $\mathbf{W}^{(s)}$ represents any matrix circularly shifted by $s$. The matrix is circularly shifted by circularly shifting each of its sub vectors. A described deterministic PCM form can be exploited to implement low-area encoders based on a forward substitution (FS) [30–34].

Since the code is quasi cyclic, the codeword can be grouped into $k_b$ information bit groups and $m_b$ parity bit groups as follows:

$$\mathbf{x} = \begin{bmatrix} \mathbf{i}_1 & \mathbf{i}_2 & \cdots & \mathbf{i}_{k_b} & \mathbf{p}_1 & \mathbf{p}_2 & \cdots & \mathbf{p}_{m_b} \end{bmatrix}. \tag{13}$$

From (5), (12), and (13), it is simple to derive the following system of equations in GF(2):

$$\sum_{l=1}^{k_b} \mathbf{h}_{1,l}\mathbf{i}_l^{\mathrm{T}} + \mathbf{p}_1^{\mathrm{T}(1)} + \mathbf{p}_2^{\mathrm{T}} = 0, \ (1^{\mathrm{st}} \text{ row})$$

$$\sum_{l=1}^{k_b} \mathbf{h}_{j,l}\mathbf{i}_l^{\mathrm{T}} + \mathbf{p}_j^{\mathrm{T}} + \mathbf{p}_{j+1}^{\mathrm{T}} = 0, \text{ for } j \in \{2,3,\ldots,r-1\} \cup \{r+1,r+2,\ldots,m_b-1\}$$

$$\sum_{l=1}^{k_b} \mathbf{h}_{r,l}\mathbf{i}_l^{\mathrm{T}} + \mathbf{p}_1^{\mathrm{T}} + \mathbf{p}_r^{\mathrm{T}} + \mathbf{p}_{r+1}^{\mathrm{T}} = 0, \ (r^{\mathrm{th}} \text{ row})$$

$$\sum_{l=1}^{k_b} \mathbf{h}_{m_b,l}\mathbf{i}_l^{\mathrm{T}} + \mathbf{p}_1^{\mathrm{T}(1)} + \mathbf{p}_{m_b}^{\mathrm{T}} = 0, \ (m_b^{\mathrm{th}} \text{ row}) \tag{14}$$

Parity group $\mathbf{p}_1$ can be calculated by adding up all the equations from (14), which gives the following expression:

$$\mathbf{p}_1^{\mathrm{T}} = \sum_{j=1}^{m_b} \sum_{l=1}^{k_b} \mathbf{h}_{j,l}\mathbf{i}_l^{\mathrm{T}}. \tag{15}$$

In hardware, each of the $m_b$ sums $\sum_{l=1}^{k_b} \mathbf{h}_{j,l}\mathbf{i}_l^{\mathrm{T}}$ is usually calculated by circularly shifting the information group vectors $\mathbf{i}_l^{\mathrm{T}}$ and by calculating the sum of shifted vectors using XOR gates. If these sums are denoted with $\lambda_j$, then the rest of the parity bit vectors can be calculated as follows:

$$\begin{aligned}
\mathbf{p}_2^{\mathrm{T}} &= \lambda_1 + \mathbf{p}_1^{\mathrm{T}(1)}; \qquad \mathbf{p}_{j+1}^{\mathrm{T}} = \lambda_j + \mathbf{p}_j^{\mathrm{T}}, \text{ for } j \in \{2,3,\ldots,r-2\}; \\
\mathbf{p}_{m_b}^{\mathrm{T}} &= \lambda_{m_b} + \mathbf{p}_1^{\mathrm{T}(1)}; \quad \mathbf{p}_j^{\mathrm{T}} = \lambda_j + \mathbf{p}_{j+1}^{\mathrm{T}}, \text{ for } j \in \{m_b-1, m_b-2,\ldots,r+1\}; \\
\mathbf{p}_r^{\mathrm{T}} &= \lambda_r + \mathbf{p}_1^{\mathrm{T}(1)} + \mathbf{p}_{r+1}^{\mathrm{T}}, \text{ if } r > 2.
\end{aligned} \tag{16}$$

An almost identical approach has been recently used for encoding 5G NR LDPC codes since their structure is essentially similar to the described structure of Wi-Fi codes [35]. Many codes from other standards have similar structural features, which can be similarly used for achieving low complexity encoding [38,39].

The disadvantages of FS-based methods are that code information should be stored in some storage memory and that there is still some sequential processing, which can limit the achievable parallelism. However, since FS is dominantly used for structured codes, the memory requirements in previously described architectures [30–35] come down to only CPM positions inside the base graph matrix and the corresponding circulant shift values, i.e., there is no need for storage of the entire PCM. Additionally, these methods do provide high flexibility, which is increasingly required in modern communication standards [1].

## 3. Encoding of 5G NR LDPC Codes

The structure of 5G NR LDPC codes is illustrated in Section 3.1. After that, Section 3.2 gives a description of the efficient encoding algorithm that is used in this paper.

### 3.1. Description of LDPC Codes in 5G NR

All 5G NR LDPC codes are quasi cyclic. As for every QC–LDPC code, the PCM consists of submatrices (CPMs) of size $Z \times Z$, which can be a circularly shifted identity matrix or a zero matrix. The general structure of a 5G NR LDPC code's PCM is

$$\mathbf{H} = \begin{bmatrix} \mathbf{A} & \mathbf{B} & 0 \\ \mathbf{C} & \mathbf{D} & \mathbf{I} \end{bmatrix}, \tag{17}$$

where $\mathbf{A}$ (of size $4Z \times k_b Z$), $\mathbf{C}$ (of size $(m_b - 4)Z \times k_b Z$) and $\mathbf{D}$ (of size $(m_b - 4)Z \times 4Z$) have conventional nonspecific quasi-cyclic structure, and $\mathbf{B}$ (of size $4Z \times 4Z$) has a form similar

to the right-hand side of IEEE 802.11n/ac/ax PCMs described in Section 2.5. Depending on the base graph, the matrix **B** can have one of the following structures:

$$\mathbf{B}_{\mathrm{BG1}} = \begin{bmatrix} \mathbf{I}^{(1)} & \mathbf{I} & 0 & 0 \\ \mathbf{I} & \mathbf{I} & \mathbf{I} & 0 \\ 0 & 0 & \mathbf{I} & \mathbf{I} \\ \mathbf{I}^{(1)} & 0 & 0 & \mathbf{I} \end{bmatrix}; \mathbf{B}_{\mathrm{BG2}} = \begin{bmatrix} \mathbf{I} & \mathbf{I} & 0 & 0 \\ 0 & \mathbf{I} & \mathbf{I} & 0 \\ \mathbf{I}^{(1)} & 0 & \mathbf{I} & \mathbf{I} \\ \mathbf{I} & 0 & 0 & \mathbf{I} \end{bmatrix}. \tag{18}$$

A scatter diagram of the base graph for both BG1 and BG2 matrices is shown in Figure 1.

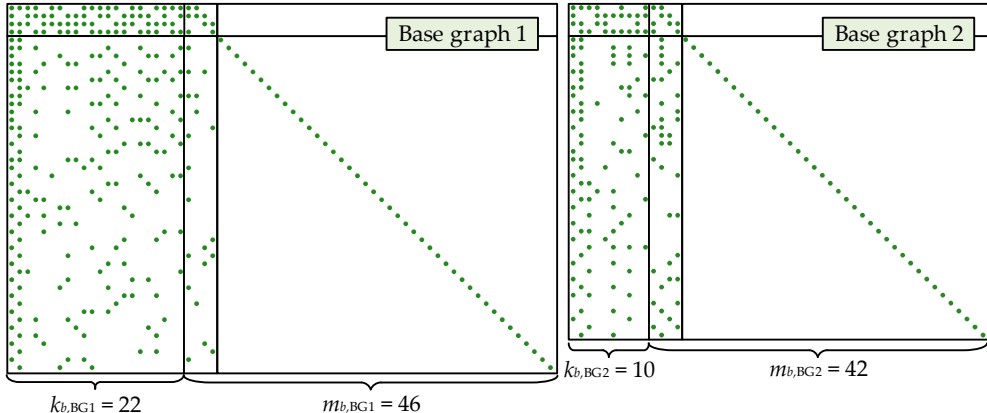

**Figure 1.** Base graph matrix structures for codes in 5G NR.

The codeword (of length $n_b Z$, where $n_b = k_b + m_b$) can be represented in systematic form as

$$\mathbf{x} = \begin{bmatrix} \mathbf{i} & \mathbf{p_c} & \mathbf{p_a} \end{bmatrix}, \tag{19}$$

where $\mathbf{p_c}$ represents a vector of so-called core parity bits and $\mathbf{p_a}$ is a vector of so-called additional parity bits [40]. The information bits vector is of length $k_b Z$, whereas the parity bit vectors are of length $4Z$ and $(m_b - 4)Z$, respectively. Since the upper right submatrix of the PCM is a zero matrix, all core parity bits can be calculated from information bits using submatrices **A** and **B**. Additional parity bits can be calculated from information and core parity bits using submatrices **C** and **D**.

The lifting size $Z$ can take values from 2 to 384 and must have a form of

$$Z = a \cdot 2^j, \tag{20}$$

where $a$ can take values from the set {2, 3, 5, 7, 9, 11, 13, 15}, which gives a total of 51 different lifting size values.

There are two base graph matrices (BG1 and BG2) available from which all codes are derived. Each base graph matrix has a corresponding set of base exponent matrices. The set includes one matrix for each possible value of parameter $a$ from (20), defined for the corresponding maximum lifting size (256, 384, 320, 224, 288, 352, 208, or 240). Every exponent matrix contains nonnegative and negative entries. Nonnegative entries represent the values by which the corresponding identity submatrices are circularly shifted. All other entries are equal to $-1$ and represent zero submatrices. The structure of a base exponent matrix is

$$\mathbf{V} = \begin{bmatrix} V_{1,1} & V_{1,2} & \cdots & V_{1,n_b} \\ V_{2,1} & V_{2,2} & \cdots & V_{1,n_b} \\ \vdots & \vdots & \ddots & \vdots \\ V_{m_b,1} & V_{m_b,2} & \cdots & V_{m_b,n_b} \end{bmatrix}. \tag{21}$$

Base exponent matrices define exponent matrices for every other lifting size whose structure is the same as

$$\mathbf{P} = \begin{bmatrix} P_{1,1} & P_{1,2} & \cdots & P_{1,n_b} \\ P_{2,1} & P_{2,2} & \cdots & P_{1,n_b} \\ \vdots & \vdots & \ddots & \vdots \\ P_{m_b,1} & P_{m_b,2} & \cdots & P_{m_b,n_b} \end{bmatrix}. \tag{22}$$

Shift values $P_{j,l}$ are calculated as follows:

$$P_{j,l} = \begin{cases} -1, & \text{if } V_{j,l} = -1 \\ V_{j,l} \bmod Z, & \text{if } V_{j,l} > -1 \end{cases}, \tag{23}$$

where mod represents modulo operation [1,5,9].

Up until today, the 5G NR standard introduces the highest number of possible LDPC codes. A practical system must have all the codes supported since the choice of which code is used is made in the runtime. Therefore, the encoder and the decoder need to be flexible, which defines a challenge for designers of practical systems.

### 3.2. Efficient Algorithm for Flexible Encoding

As mentioned earlier, the 5G NR communication standard imposes the necessity for high LDPC encoder flexibility. There are a total of 102 codes derived just from two base graph matrices by lifting each with every one of 51 lifting sizes. The standard also introduces the shortening of base graph matrices, which additionally increases the number of possible codes by tens of times [1]. This flexibility cannot be achieved by any method that requires either multiplying with generator matrix or any matrix inversion without extremely high usage of memory or logic resources. Therefore, the flexible encoding must be achieved based on the previously described forward substitution method.

The forward substitution-based encoding for BG1 codes has been presented in [35]. It has been shown that this method is multiple times more efficient than straightforward and RU methods and that fast encoding can be achieved.

The first step is the calculation of core parity bits. After that, the additional parity bits can be calculated using only the lower part of the PCM. Similar to Wi-Fi codes, the encoding process solves the following system of equations:

$$\mathbf{H}\mathbf{x}^{\mathrm{T}} = 0 \Rightarrow \begin{bmatrix} \mathbf{A} & \mathbf{B} & 0 \\ \mathbf{C} & \mathbf{D} & \mathbf{I} \end{bmatrix} \begin{bmatrix} \mathbf{i}^{\mathrm{T}} \\ \mathbf{p}_{\mathbf{c}}^{\mathrm{T}} \\ \mathbf{p}_{\mathbf{a}}^{\mathrm{T}} \end{bmatrix} = 0. \tag{24}$$

This system can be split into the smaller system of equations from which core parity bits can be calculated (25) and the set of equations from which additional parity bits are afterward calculated (26).

$$\begin{bmatrix} \mathbf{a}_{1,1} & \mathbf{a}_{1,2} & \cdots & \mathbf{a}_{1,k_b} \\ \mathbf{a}_{2,1} & \mathbf{a}_{2,2} & \cdots & \mathbf{a}_{2,k_b} \\ \mathbf{a}_{3,1} & \mathbf{a}_{3,2} & \cdots & \mathbf{a}_{3,k_b} \\ \mathbf{a}_{4,1} & \mathbf{a}_{4,2} & \cdots & \mathbf{a}_{4,k_b} \end{bmatrix} \times \begin{bmatrix} \mathbf{i}_1^{\mathrm{T}} \\ \mathbf{i}_2^{\mathrm{T}} \\ \vdots \\ \mathbf{i}_{k_b}^{\mathrm{T}} \end{bmatrix} + \begin{bmatrix} \mathbf{b}_{1,1} & \mathbf{b}_{1,2} & \mathbf{b}_{1,3} & \mathbf{b}_{1,4} \\ \mathbf{b}_{2,1} & \mathbf{b}_{2,2} & \mathbf{b}_{2,3} & \mathbf{b}_{2,4} \\ \mathbf{b}_{3,1} & \mathbf{b}_{3,2} & \mathbf{b}_{3,3} & \mathbf{b}_{3,4} \\ \mathbf{b}_{4,1} & \mathbf{b}_{4,2} & \mathbf{b}_{4,3} & \mathbf{b}_{4,4} \end{bmatrix} \times \begin{bmatrix} \mathbf{p}_{\mathbf{c},1}^{\mathrm{T}} \\ \mathbf{p}_{\mathbf{c},2}^{\mathrm{T}} \\ \mathbf{p}_{\mathbf{c},3}^{\mathrm{T}} \\ \mathbf{p}_{\mathbf{c},4}^{\mathrm{T}} \end{bmatrix} = 0, \tag{25}$$

$$\mathbf{p}_{\mathbf{a}}^{\mathrm{T}} = \begin{bmatrix} \mathbf{c}_{1,1} & \mathbf{c}_{1,2} & \cdots & \mathbf{c}_{1,k_b} & \mathbf{d}_{1,1} & \mathbf{d}_{1,2} & \mathbf{d}_{1,3} & \mathbf{d}_{1,4} \\ \mathbf{c}_{2,1} & \mathbf{c}_{2,2} & \cdots & \mathbf{c}_{2,k_b} & \mathbf{d}_{2,1} & \mathbf{d}_{2,2} & \mathbf{d}_{2,3} & \mathbf{d}_{2,4} \\ \vdots & \vdots & \ddots & \vdots & \vdots & \vdots & \vdots & \vdots \\ \mathbf{c}_{m_b-4,1} & \mathbf{c}_{m_b-4,2} & \cdots & \mathbf{c}_{m_b-4,k_b} & \mathbf{d}_{m_b-4,1} & \mathbf{d}_{m_b-4,2} & \mathbf{d}_{m_b-4,3} & \mathbf{d}_{m_b-4,4} \end{bmatrix} \times \begin{bmatrix} \mathbf{i}^{\mathrm{T}} \\ \mathbf{p}_{\mathbf{c}}^{\mathrm{T}} \end{bmatrix}. \tag{26}$$

For BG1 codes, the system (25) is just a special case of the system (12), where $r = 2$. Therefore, if the intermediate results are denoted as

$$\boldsymbol{\lambda}_j^{\mathrm{T}} = \sum_{l=1}^{k_b} \mathbf{a}_{j,l} \mathbf{i}_l^{\mathrm{T}}, \; j \in \{1, 2, 3, 4\}, \tag{27}$$

core parity bits can be calculated using the following expressions:

$$\mathbf{p}_{\mathbf{c},1} = \sum_{j=1}^{4} \boldsymbol{\lambda}_j; \quad \mathbf{p}_{\mathbf{c},2} = \boldsymbol{\lambda}_1 + \mathbf{p}_{\mathbf{c},1}^{(1)}; \quad \mathbf{p}_{\mathbf{c},4} = \boldsymbol{\lambda}_4 + \mathbf{p}_{\mathbf{c},1}^{(1)}; \quad \mathbf{p}_{\mathbf{c},3} = \boldsymbol{\lambda}_3 + \mathbf{p}_{\mathbf{c},4} \; . \tag{28}$$

In [35], the system (25) is solved only for BG1 codes. In this paper, flexibility was a key requirement. Therefore, the analysis is expanded to codes defined by BG2 too. For BG2, the system (25) can be written as follows:

$$
\begin{aligned}
\sum_{l=1}^{k_b} \mathbf{a}_{1,l} \mathbf{i}_l^{\mathrm{T}} + \mathbf{p}_{\mathbf{c},1}^{\mathrm{T}} + \mathbf{p}_{\mathbf{c},2}^{\mathrm{T}} &= 0, \\
\sum_{l=1}^{k_b} \mathbf{a}_{2,l} \mathbf{i}_l^{\mathrm{T}} + \mathbf{p}_{\mathbf{c},2}^{\mathrm{T}} + \mathbf{p}_{\mathbf{c},3}^{\mathrm{T}} &= 0, \\
\sum_{l=1}^{k_b} \mathbf{a}_{3,l} \mathbf{i}_l^{\mathrm{T}} + \mathbf{p}_{\mathbf{c},1}^{\mathrm{T}(1)} + \mathbf{p}_{\mathbf{c},3}^{\mathrm{T}} + \mathbf{p}_{\mathbf{c},4}^{\mathrm{T}} &= 0, \\
\sum_{l=1}^{k_b} \mathbf{a}_{4,l} \mathbf{i}_l^{\mathrm{T}} + \mathbf{p}_{\mathbf{c},1}^{\mathrm{T}} + \mathbf{p}_{\mathbf{c},4}^{\mathrm{T}} &= 0.
\end{aligned}
\tag{29}
$$

By adding up all the above equations, it is simple to obtain the first core parity bit group circularly shifted by 1 position as

$$\mathbf{p}_{\mathbf{c},1}^{(1)} = \sum_{j=1}^{4} \boldsymbol{\lambda}_j. \tag{30}$$

This vector should be shifted back by 1 to obtain the first core parity bit group in the appropriate arrangement. With the first group available, all other groups are directly calculated from equations in (29) as follows:

$$\mathbf{p}_{\mathbf{c},2} = \boldsymbol{\lambda}_1 + \mathbf{p}_{\mathbf{c},1}; \quad \mathbf{p}_{\mathbf{c},3} = \boldsymbol{\lambda}_2 + \mathbf{p}_{\mathbf{c},2}; \quad \mathbf{p}_{\mathbf{c},4} = \boldsymbol{\lambda}_4 + \mathbf{p}_{\mathbf{c},1} \; . \tag{31}$$

The described algorithm is used in this paper for efficient hardware implementation.

## 4. Optimal Encoding Schedules in Partially Parallel Encoding

The encoding algorithm described in Section 3.2 can be implemented with various levels of parallelism. Fully parallel implementation would require extremely high logic usage and a relatively long critical path since the codeword can be up to 26,112 bits long and since flexibility is one of the main requirements in 5G NR. Partially parallel implementations can provide a compromise between achievable throughput and utilization of hardware resources. This section describes various approaches for partially parallel encoding and analyses corresponding hardware usage utilization for each of the approaches.

### 4.1. Partially Parallel Processing in 5G NR LDPC Encoding

The encoding process mainly consists of circular shifting Z-bit vectors and calculating the XOR operation of all the shifted values. In partially parallel processing, the hardware that is used for shifting, later in the text called circular shifter (CS), must be generic, i.e., it needs to support all shift values and all shift (lifting) sizes that are defined by the standard. A requirement for the flexibility of CSs drastically increases their hardware resources requirements to the point where shifters take the most hardware resources in the entire

encoder. Therefore, if the hardware usage efficiency is the key requirement, it is important that these components are utilized as much as possible, i.e., that they are rarely idle.

Partially parallel processing can be conducted in multiple ways. If only one CS is used, the encoding is performed serially by processing one CPM at a time. Note that this encoding is still partially parallel since the CS takes the entire vector of $Z$ bits. Figure 2 shows such architecture's encoding schedule. The schedule is shown for BG1 codes since BG2 codes have a similar structure but a smaller number of CPMs in the PCM. In Figure 2, numbers at CPM positions represent ordinals of clock cycles in which the corresponding CPMs are processed. In the beginning, the encoder takes information bit groups that correspond to the first four base graph rows necessary for the calculation of vectors $\lambda_1$, $\lambda_2$, $\lambda_3$, and $\lambda_4$. Based on calculated $\lambda$ vectors, core parity bits are computed as in (28) or as in (30) and (31). This can be performed in parallel with further information bit groups shifting for later rows and their XOR sum calculation. Due to this parallel computation, all CPMs that are used for core parity bits calculation, based on vectors $\lambda_1$, $\lambda_2$, $\lambda_3$, and $\lambda_4$, are placed in the same clock cycle in Figure 2.

| | 1 | 2 | 3 | 4 | 5 | 6 | 7 | 8 | 9 | 10 | 11 | 12 | 13 | 14 | 15 | 16 | 17 | 18 | 19 | 20 | 21 | 22 | 23 | 24 | 25 | 26 | 27 | 28 | 29 | 30 | 31 | | 67 | 68 |
|---|---|---|---|---|---|---|---|---|---|---|---|---|---|---|---|---|---|---|---|---|---|---|---|---|---|---|---|---|---|---|---|---|---|---|
| 1 | 1 | 2 | 3 | 4 | | 5 | | 6 | | | 7 | 8 | 9 | 10 | 11 | | 12 | 13 | | 14 | 15 | 16 | 17 | | | | | | | | | ⋯ | | |
| 2 | | | 18 | 19 | 20 | 21 | | 22 | 23 | 24 | | 25 | 26 | | 27 | 28 | 29 | 30 | | 31 | | 32 | | 67 | | | | | | | | | | |
| 3 | 33 | 34 | 35 | | 36 | 37 | 38 | 39 | 40 | 41 | 42 | | | 43 | 44 | 45 | | 46 | 47 | 48 | 49 | | | | | | | | | | | | | |
| 4 | 50 | 51 | | 52 | 53 | | | 54 | 55 | 56 | | | 57 | 58 | 59 | 60 | 61 | | 62 | 63 | 64 | | 65 | 66 | | | | | | | | | | |
| 5 | 67 | 68 | | | | | | | | | | | | | | | | | | | | | | | | | | | | | | | | |
| 6 | 69 | 70 | | 71 | | | | | | | | | 72 | | | 73 | | | | | 74 | 75 | | | | | | | | | | | | |
| 7 | 76 | | | | | 77 | | | | | 78 | 79 | | 80 | | | | 81 | 82 | | 83 | | | | | | | | | | | | | |
| 8 | 84 | 85 | | | 86 | | | 87 | 88 | | | | | 89 | | | | | | | | | | | | | | | | | | | | |
| 9 | 90 | 91 | | 92 | | | | | | | | | 93 | | | | 94 | | | 94 | | 96 | 97 | | 98 | | | | | | | ⋯ | | |
| ⋮ | | | | | | | | | | | | | | | | | | | | | | | | | | | | | | | | | | |
| 45 | 259 | | | | | | | 260 | | 261 | | | | | | | | | | | | 262 | | | | | | | | | | ⋯ | | |
| 46 | | 263 | | | | | 264 | | | | 265 | | | | | | | | | | | | | | | | | | | | | | | |

**Figure 2.** Encoding schedule for BG1 codes when serially processing single CPM at a time. Numbers represent the ordinal of the clock cycle in which the corresponding CPM is processed.

The serial processing schedule requires 265 clock periods for all parity bits generation for BG1 codes and 150 clock periods for BG2 codes. Although the throughput of such an approach is the smallest of all partially parallel schedules, the HUE is the highest since only one CS is used and is never idle.

A similar schedule has been presented in [32] for Wi-Fi codes, but its order of processing was column oriented, meaning that the processing was performed column by column with the storage of intermediate results in local memory.

The encoding throughput can be increased if a larger number of parallel processing units is used, i.e., by exploiting higher parallelism. Figure 3 shows the encoding schedule proposed in [34] for Wi-Fi codes but here applied to 5G NR codes. All CPMs that belong to a single column are processed in parallel. After the first $k_b$ columns are processed, all $\lambda$ vectors are ready and core parity bits can be calculated. It is convenient to wait at least one clock period for core parity bits to be ready before the remaining four columns are processed. This is carried out because of the pipelining, which makes the critical path much shorter. The schedule requires at least 27 ($k_{b,\text{BG1}} + 1 + 4$) clock periods for all parity bits generation for BG1 codes and at least 15 ($k_{b,\text{BG2}} + 1 + 4$) clock periods for BG2 codes. The described method requires a large number of CSs: 46 for BG1 codes and 42 for BG2 codes. Since the encoder should be flexible, it is necessary to use a larger of these two numbers.

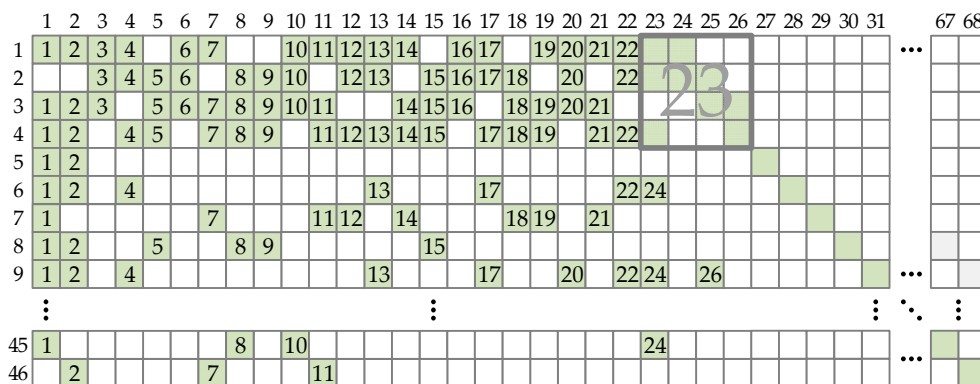

**Figure 3.** Encoding schedule for BG1 codes when processing the entire base graph matrix column at the same time. Numbers represent the ordinal of the clock cycle in which the corresponding CPM is processed.

In order to reduce used hardware resources, a row-based parallel encoding can be executed [31,35]. Such a schedule is shown in Figure 4. All CPMs from a single row are processed in a single clock period. Instead of shifting one bit group at a time, this schedule requires that all information bits, and later core parity bits, are available in registers. Consequently, it is often required to wait for information bits to be written to registers from the serial input stream [35]. However, assuming that the double buffering is available at the data input, this stall time can be avoided. The GF(2) sum of shifted vectors should be calculated using XOR gates placed in a tree-like structure.

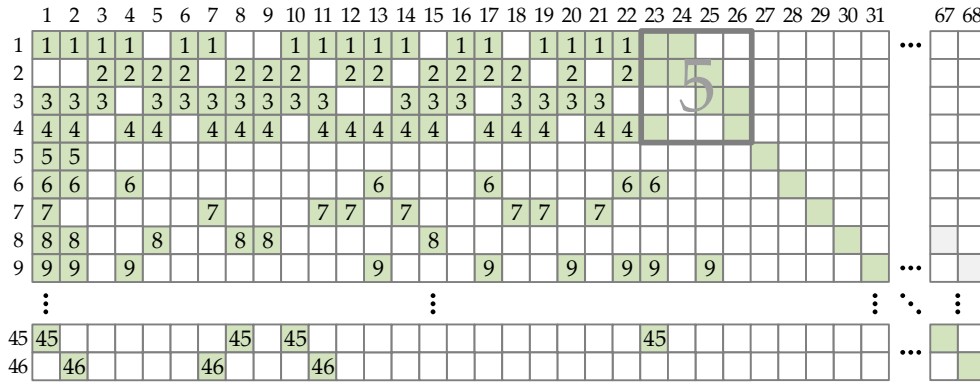

**Figure 4.** Encoding schedule for BG1 codes when processing the entire base graph matrix row at the same time. Numbers represent the ordinal of the clock cycle in which the corresponding CPM is processed.

The required number of clock cycles for one codeword encoding is at least 46 ($m_{b,\mathrm{BG1}}$) for BG1 codes and at least 42 ($m_{b,\mathrm{BG2}}$) for BG2 codes, which is significantly higher when compared with the column-based parallel encoding. However, the required number of CSs is reduced: 26 for BG1 codes and 14 for BG2 codes [35]. Again, it is necessary to use the larger of these two numbers because of the flexibility.

The encoder information throughput can be calculated as follows:

$$Thr = \frac{k_b Z f_{CLK}}{N_{CPC}}. \tag{32}$$

where $f_{CLK}$ is the encoder's operating frequency, and $N_{CPC}$ is the number of clock cycles necessary for a single codeword encoding –clocks per codeword. Since CSs take the largest amount of logic resources, hardware usage can be approximately expressed in the number

of CSs. For a better comparison of approximate hardware usage efficiencies for described encoding schedules, a circular shifter's efficiency is defined as follows:

$$Eff_{CS} = \frac{Thr}{N_{CS}},\tag{33}$$

where $N_{cs}$ is the number of used CSs. This parameter values calculated for the longest codewords ($k_{max,BG1} = k_{b,BG1}Z_{max} = 8448$, and $k_{max,BG2} = k_{b,BG2}Z_{max} = 3840$) are summarized in Table 1. As can be inferred from the results, the most efficient schedule is serial processing, although it gives the lowest throughput. Highly parallel encoding schedules are less efficient because a large number of shifters are unused, which is a consequence of the fact that the base graph matrix's rows and columns are not fully filled.

**Table 1.** Comparison of approximated hardware usage efficiencies of a flexible encoder for various encoding schedules and for the longest codeword ($Z = 384$) in b/cycle per circular shifter used.

| Encoding Schedule | $N_{CPC}$ | | $N_{CS}$ | $Eff_{CS}$ (b/Cycle per CS) | |
|---|---|---|---|---|---|
| | BG1 | BG2 | | BG1 | BG2 |
| Serial | 265 | 150 | 1 | 31.9 | 25.6 |
| Column based parallel | 27 | 15 | 46 | 6.8 | 5.6 |
| Row based parallel | 46 | 42 | 26 | 7.1 | 3.5 |

For the reasons of efficiency, in this paper, serial CPM processing is implemented for codes with high lifting size ($Z > 192$). Only one CS is used. However, the CS is designed in such a way that it can be reconfigured to reuse logic resources and to work as two or more independent CSs if codes with lower lifting size values are used. The CS design and the entire encoder architecture are described in detail in Section 5.

Since more than one CS is available for shorter codes, the serial encoding schedule can be then parallelized to increase the HUE. The proposed encoding schedule processes more than one row at a time. Its example of a situation when four shifters are available is shown in Figure 5. The encoder takes information bit groups, shifts each of them, and calculates intermediate results for all $\lambda$ vectors. During the next four rows processing, core parity bits are calculated and prepared for calculation of additional parity bits.

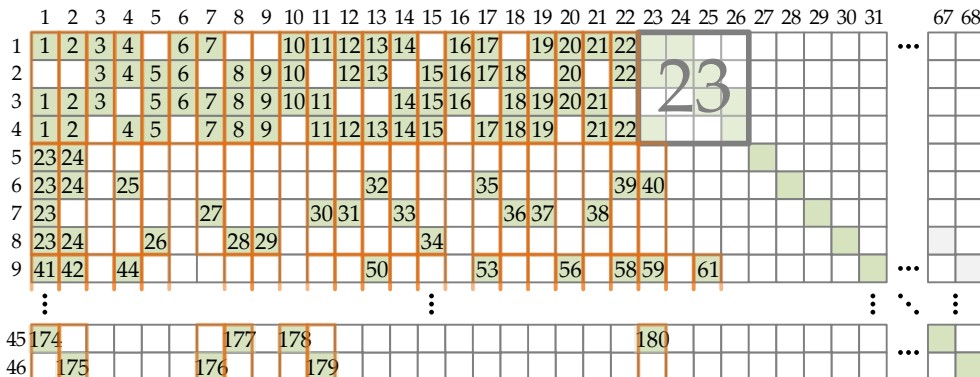

**Figure 5.** Proposed encoding schedule for BG1 codes when four CSs are available. Numbers represent the ordinal of the clock cycle in which the corresponding CPM is processed.

Using the described multirow serial schedule with four CSs, the encoding time can be reduced to 180 clock cycles for BG1 codes and 94 clock cycles for BG2 codes. However, during encoding, even three out of four available CSs are frequently idle due to the PCM's structure. This is easily noticeable in Figure 5 for all rows used in additional parity bits calculation. The reason for such PCM design is an attempt to avoid pipeline conflicts

that happen in layered decoding [41] architectures because of read-after-write (RAW) data hazards [5,11].

Although PCM has a structure designed to support decoder efficiency, it can be processed in a way that is optimal for the encoder. For example, the sixth-row group shown in Figure 5 uses almost the same information and core parity bits as the ninth-row group. A significantly higher efficiency would be obtained if the sixth and ninth rows are processed together. Consequently, instead of conventionally processing adjacent rows groups in parallel, the encoding can be accomplished by processing the rows that use the same or at least almost the same information and core parity bits. Therefore, the proposed encoding schedule can be optimized for better hardware efficiency. A method for obtaining the optimal schedule is described in Section 4.2.

### 4.2. Multirow Serial Encoding Schedule Optimization

Finding the optimal processing order belongs to the traveling salesman problem class and can be efficiently solved using a genetic algorithm (GA) [42,43]. The GA has been previously used for finding the optimal layered decoding computation schedule [11,43]. The goal was to find the processing schedule that would provide the smallest number of pipeline RAW data hazards. A schedule that provides the smallest data hazards number should have a minimal data dependency between the adjacent rows. Therefore, the optimization procedure results in the schedule that has adjacent rows whose CPMs are in the different columns.

However, in this work, the schedule is optimized in a completely opposite manner; the cost function is written to favor those processing orders whose adjacent rows have as many as possible CPMs in the same columns. The result of the cost function returns the number of clock cycles that are needed for encoding completion, and it is exclusively dependent on the processing order. This criterion should result in a schedule that maximizes the available CSs utilization and therefore gives the highest HUE.

In Section 4.1, all core parity bits are treated in the same way as information bits in the process of additional bits calculation. Whenever any core parity bits vector participates in the GF(2) sum for calculation of additional parity bits, the encoder should allocate a separate clock cycle for each of the core parity groups, as shown in Figure 5. However, if there are free CSs available, it is possible to use them to circularly shift core parity bits in parallel with the information bits shifting. In this way, there is no need for additional clock cycles used for processing of core parity-related CPMs, and hence, the throughput is increased. The illustration of repositioning the CPMs processing is shown in Figure 6. The described repositioning is included in the optimization process.

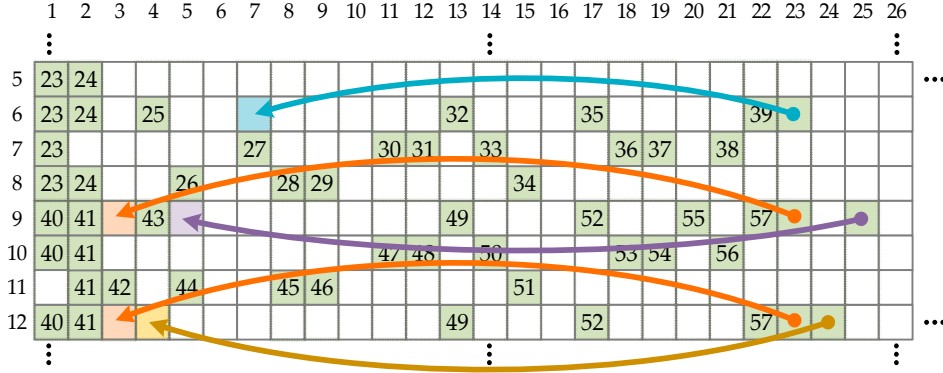

**Figure 6.** Illustration of processing repositioning for core parity bits related CPMs.

Finally, it is important to mention that the results of the first four rows in the base graph matrix are always calculated in the beginning since the core parity bits are necessary for later calculation. Reordering is performed only for the processing of rows used for additional parity bits calculation.

The GA optimization consists of the following phases: (1) generation of the initial population, after which the later phases are iteratively repeated; (2) selection of the best individuals; (3) crossover of the selected individuals; and (4) mutation. The initial population contains a large set of vectors, which represent the rows processing order. Each individual is a random permutation of the original processing order, which is a vector of natural numbers valued from 5 to $m_b$. After the population is generated, the set of processing orders that give the smallest numbers of encoding clock-cycles is selected for reproduction.

The illustration of the crossover of two vectors (parents) is shown in Figure 7. For the sake of simplicity, the example from Figure 7 uses individuals generated from a vector of natural numbers from 1 to 20, but it demonstrates a procedure applicable to any vectors. A subvector is cut from the first parent at random positions, after which it is placed in the child vector at the same position where it stood in the parent vector. The remainder of the child vector is filled with elements from the second parent vector while keeping the original order of elements. Those elements which are the same as already placed elements originating from the first vector are skipped. This is carried out to keep the uniqueness of each element in a vector. The crossover is repeated until a new population is formed. A mutation is performed by randomly switching two elements of some randomly selected vectors. The newly generated population is used for another selection, and the process is repeated until the maximal number of generations is reached.

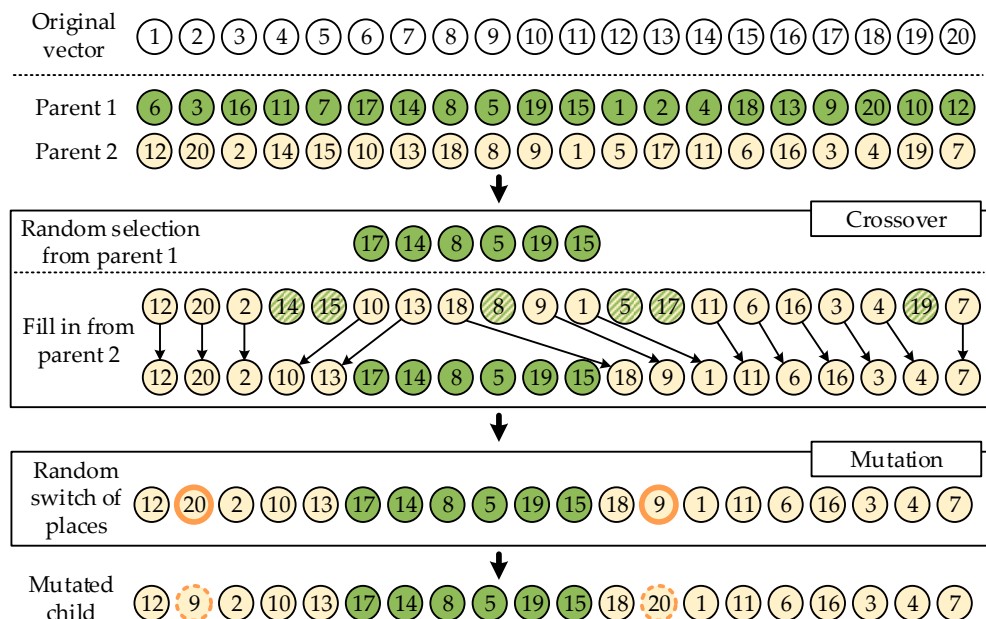

**Figure 7.** Genetic algorithm crossover and mutation examples for 20-element vectors.

## 5. Proposed Hardware Realization of Flexible 5G NR LDPC Encoder

This section presents the hardware implementation of the proposed encoding algorithm and scheduling. The flexible scheduling described in Section 4 requires a thorough design of the circular shifting network. Therefore, in Section 5.1, the proposed design for such a shifting network is presented. The architecture of the proposed encoder is presented in Section 5.2.

### 5.1. Circular Shifting Network for High Flexibility

The circular shifting network is designed to work in one of the following three modes: (1) as a single CS when lifting size is in the range $192 < Z \leq 384$, (2) as two independent CSs when lifting size is in the range $96 < Z \leq 192$, and (3) as four independent CSs when lifting size is in the range $2 \leq Z \leq 96$. For supporting such functionality, the chosen basic component for the entire network is a CS that supports all 5G NR lifting sizes of up to 96

(CS96). Four CS96 components are used as either independent CSs or as partial CSs for higher lifting sizes.

A CS96 shifter is designed as a two-stage shifter as in [11,44], where the first stage is a pre-rotator network and the second stage is a QSN (abbreviated from QC–LDPC Shift Network) circular shifter [45]. As stated in (20), the 5G NR lifting size can take values of the form $Z = a \cdot 2^j$, where $j$ goes from 0 for smallest to 7 for largest lifting sizes. For lifting sizes supported by the CS96 shifter, $j$ is always less than or equal to 5. Given that $j$ can take values as small as 0, the pre-rotator network must have outputs for multiple rotation sizes [11], which is obtained as in [46].

With CS96 available, CS for higher lifting sizes is realized using appropriate bit permutations and reordering before and after CS96 blocks. A rationale for permutation and reordering network design is based on the possibility of rearranging CPMs to a group of smaller submatrices of which some are also circularly shifted identity matrices of size $Z/D \times Z/D$, where $D$ is a natural number. Such rearranging has been previously used for the design of LDPC decoders that support lower parallelism than $Z$ and for preventing pipeline RAW data hazards [43], but here, it is applied to the design of CS. The CPM splitting relies on row and column permutations. Both row and column indexes in the new matrix are calculated using the following formula:

$$idx_{new} = (idx_{old} \bmod D) \cdot \frac{Z}{D} + \left\lfloor \frac{idx_{old}}{D} \right\rfloor. \tag{34}$$

Examples of presented permutations for $D = 2$ and $D = 4$ are shown in Figure 8. For the sake of simplicity and clearer presentation, the original lifting size is 16. A generalized illustration of CPM splitting for any lifting size is shown in Figure 9. One exponent matrix entry $P$ gives $2 \times 2 = 4$ new entries if $D = 2$, and $4 \times 4 = 16$ new entries if $D = 4$.

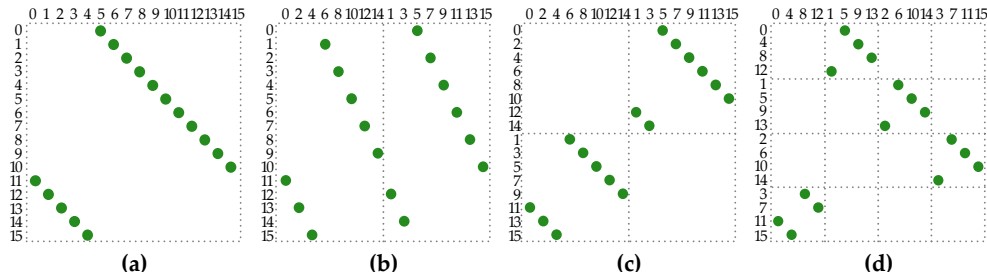

**Figure 8.** CPM splitting examples: (**a**) the original CPM, (**b**) original CPM with columns permuted for $D = 2$, (**c**) final permuted CPM for $D = 2$, and (**d**) final permuted CPM for $D = 4$.

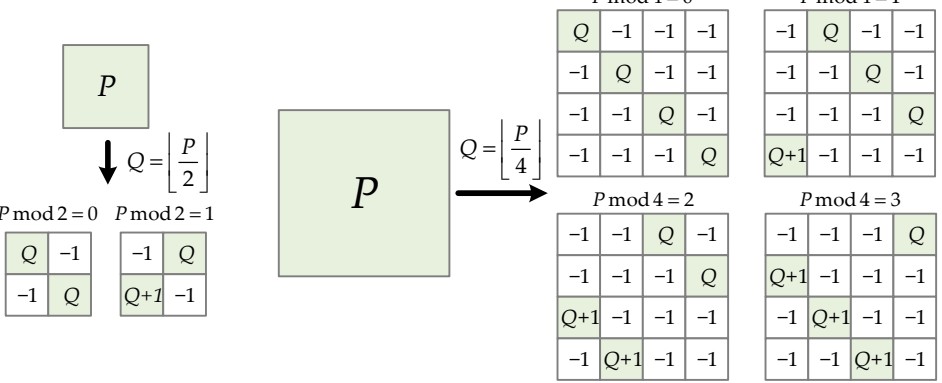

**Figure 9.** General circulant permutation matrix splitting results for $D = 2$ and $D = 4$. $P$ is the shift value from the exponent matrix. Exponent matrix entry value $-1$ represents a zero submatrix.

The described property proves that a single circular shift for a large lifting size can be split into multiple circular shifts for a smaller lifting size. Therefore, the circular shifting network for large lifting sizes can be designed by using independent smaller lifting sizes CSs. This can be performed using the following steps. Input bits are firstly permuted using the relation from (34) to obtain independent groups of data. Each group of data is circularly shifted in shifters of smaller lifting sizes. The shifted data are in the permuted order and hence need an inverse permutation to obtain the proper output bits arrangement. The CS architecture that implements this method suitable for 5G NR codes is shown in Figure 10. In addition to the already mentioned permutations, the shifting network needs a group reordering stage, which depends on the remainder after the division of the original shift value with $D$. For example, for $D = 4$, if the remainder is 0, there is no need for reordering, which can be concluded from positions of newly generated CPMs in Figure 9. However, if the remainder is 1, the first bit group must be placed last, the second must be placed first, the third must be placed second, the fourth must be placed third, etc.

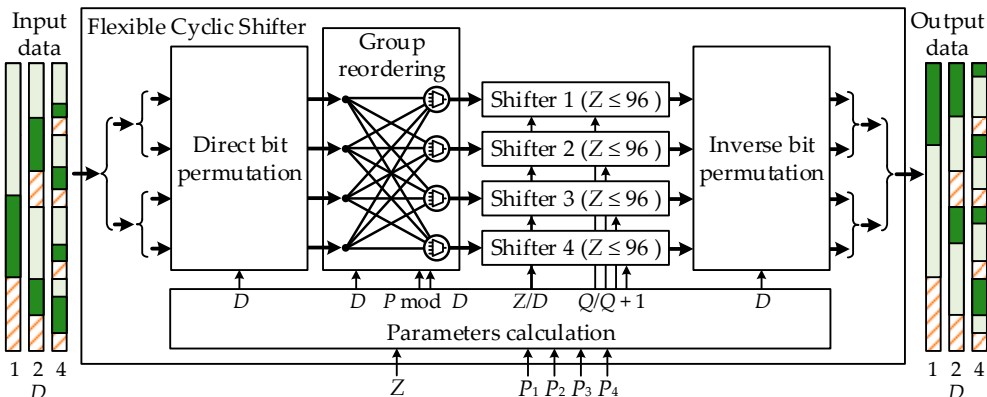

**Figure 10.** Shifting network architecture for the proposed 5G NR encoder. The network is capable of circular shifting four independent groups of data if $Z \leq 96$, two independent groups of data if $96 < Z \leq 192$, and one group of data if $192 < Z \leq 384$.

The described architecture provides another important feature. The shifted bits for lifting sizes other than 96, 192, or 384 are aligned to one side of the data block. Several examples are shown in Figure 10. Light green and dark green areas represent groups of $Z$ valid bits that should be shifted, whereas the hatched area represents nonvalid data. This property is of crucial importance since such data alignment drastically simplifies the remaining encoder blocks: core parity bits calculator, input data interface, and output data interface.

Finally, the proposed procedure can be easily applied to a more generic CS network that can shift more than four independent blocks of data. However, in such cases, due to the high irregularity of check nodes in 5G NR LDPC codes, in this particular application, the ratio of the throughput gain with respect to the additional complexity would not be as high as it is for the former cases. Additionally, the processing of the first four rows in the base graph matrix cannot be further parallelized since core parity bits are necessary for later processing. Nevertheless, other LDPC encoder or decoder architecture may have more benefits from additional shifting network splitting. For example, in order to obtain better HUE, row-based parallel architectures could use a shifter that processes all inputs in parallel for small lifting size values but combine parallel and serial processing for higher lifting sizes. Again, for 5G NR, PCM has a broad range of row weight values, and such a shifter would not always be used in its full capacity, but for codes that have relatively high row weights for all rows, not just for the first $4Z$ as in 5G NR, benefits to the HUE would be significant. Similar conclusions apply to codes whose column regularity is higher.

### 5.2. LDPC Encoder Architecture

The hardware architecture of the proposed flexible encoder is shown in Figure 11. Information bits are stored in the input buffer dual-port memory. In order to implement double buffering, the input buffer can receive information bits for two adjacent codewords. One half of memory is used for reads during the current codeword encoding, while the other half is used for information bits storage for the next codeword. Consequently, there is no need for any information bits repackaging when the encoding starts, which removes the need for additional clock cycles and increases the encoding speed.

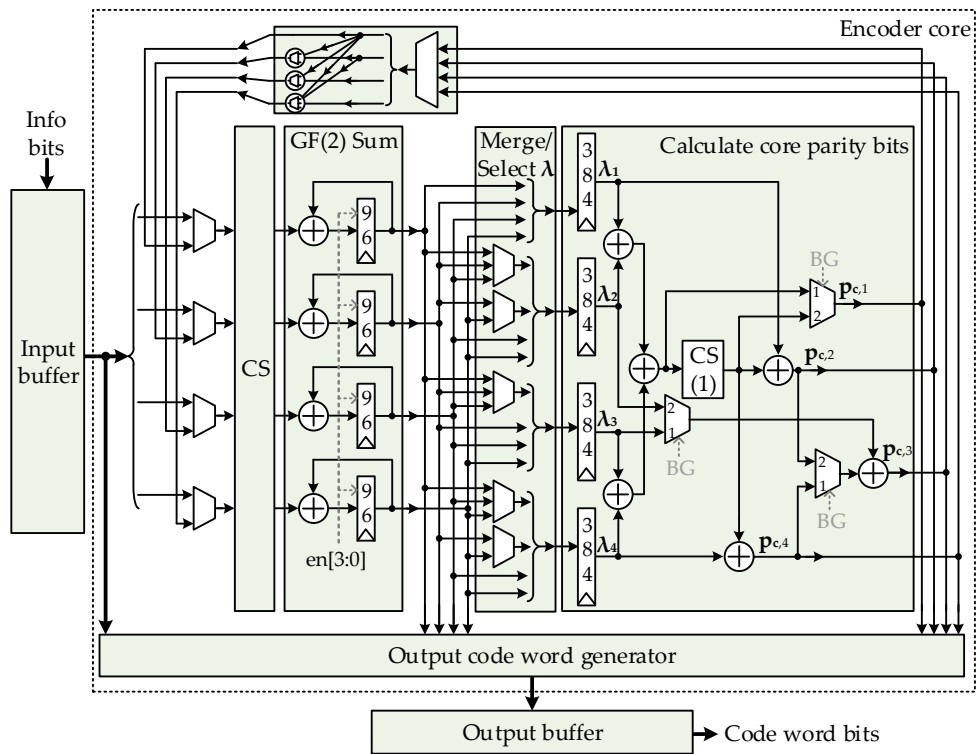

**Figure 11.** Proposed 5G NR LDPC encoder architecture.

Depending on the lifting size, the encoder can work in one of the three modes determined by the circular shifter. If lifting size is in the range $192 < Z \leq 384$, the encoder serially processes one CPM per clock cycle, from one row of the base graph matrix. The information bits group of maximum bit length 384 is read from the input buffer. Info bits are shifted and used for the calculation of GF(2) sums for $\lambda$ vectors. All GF(2) additions are realized using the XOR operation. If lifting size is in the range $96 < Z \leq 192$, the encoder serially processes up to two CPMs per clock cycle, from two adjacent rows defined by the optimized schedule. In this case, a word that is read from the input buffer memory consists of an information bits group (of maximum bit length 192), and its copy is placed in the word's high bits. This is provided by the input interface, which is not shown in Figure 11. Circular shifting network works as two independent CSs with shift sizes equal to the given code lifting size. If lifting size is in the range $2 \leq Z \leq 96$, the encoder serially processes up to four CPMs per clock cycle, from four adjacent rows defined by the corresponding optimized schedule. The input buffer now contains words that consist of four identical copies of information bits groups. They are independently shifted in the CS and then used for the calculation of four independent GF(2) sums. During the process of $\lambda$ vectors calculation, information bits are also passed to the output codeword generator that writes the data to the output buffer.

Since, as described, the parallelism of the $\lambda$ vectors calculation is different for different lifting sizes, there is a need for a separate block, marked as "Merge/Select $\lambda$" in Figure 11, whicharranges the $\lambda$ vectors data in the appropriate order for later core parity bits calculation.

After the calculation of all $\lambda$ vectors, the process continues in the same manner for the next rows in the base graph matrix. During that time, core parity bits are calculated in parallel. BG1 and BG2 have different equations for the calculation of core parity bits. However, most of the logic can be shared by inserting a few additional multiplexers that rearrange data paths depending on the base graph selected. The circular shifter marked as CS(1) shifts data only by 1, to the left or right depending on the base graph. Since the shifter should support only two shift values, it can be implemented using just a fragment of resources that are used for the main CS, although it should still support all lifting sizes defined in the 5G NR standard. When calculated, the core parity bits are also passed to the output codeword generator.

Calculated core parity bits are returned to the main CS input since they are used for additional parity bits calculation. One of the four core parity bit groups is selected based on the control CPM data. Depending on the lifting size, the selected core parity bits group is copied in the same way as it has been for information bits in the input buffer. Finally, four sets of multiplexers determine whether the information bits group or the core parity bits group should be passed to the CS. Outputs of the "GF(2) Sum" block are now additional parity bits, which are passed to the output codeword generator.

The entire architecture is pipelined to obtain high operating frequency. Therefore, it is necessary that the encoding schedule does not use core parity bits for the calculation of additional parity bits until the former are ready. Ideally, the first rows that are used for additional parity bits calculation should not contain CPMs in the columns related to the core parity bits. This is provided by giving the additional constraint to the processing order optimization.

In the end, it should be mentioned that almost identical architecture could be applied for encoding of IEEE 802.16e (WiMAX) [38] LPDC codes. There are two main differences. First is that WiMAX lifting sizes are between 24 and 96. Therefore, the main CS would be designed using four CS24 components, instead of CS96s. The second difference is based on the fact that WiMAX has only core parity bits which can be calculated recursively in a way similar to (15) and (16). The proposed 5G NR encoder calculates all core parity bits in parallel, but for WiMAX, it may be more efficient to conduct that serially because the number of parity bits is not the same for all codes. Additional CS needed for parity bits calculation should support slightly more than only two shift values as it is in the 5G NR encoder.

## 6. Results and Discussion

### 6.1. Encoding Schedule Optimization Results

Optimization of the processing order provided a strong decrease in the encoding time. Examples of optimized encoding schedules are shown in Figure 12. Examples are shown for a processing order when four CSs are available. It is obvious that rows are grouped in a way that the same information bits are used for multiple CPMs at the same time. This results in a significant increase in the encoding speed. Moreover, the core parity bits processing is merged with the processing of some information bits when free CSs are available. This repositioning is of great importance for the additional encoding speed increase.

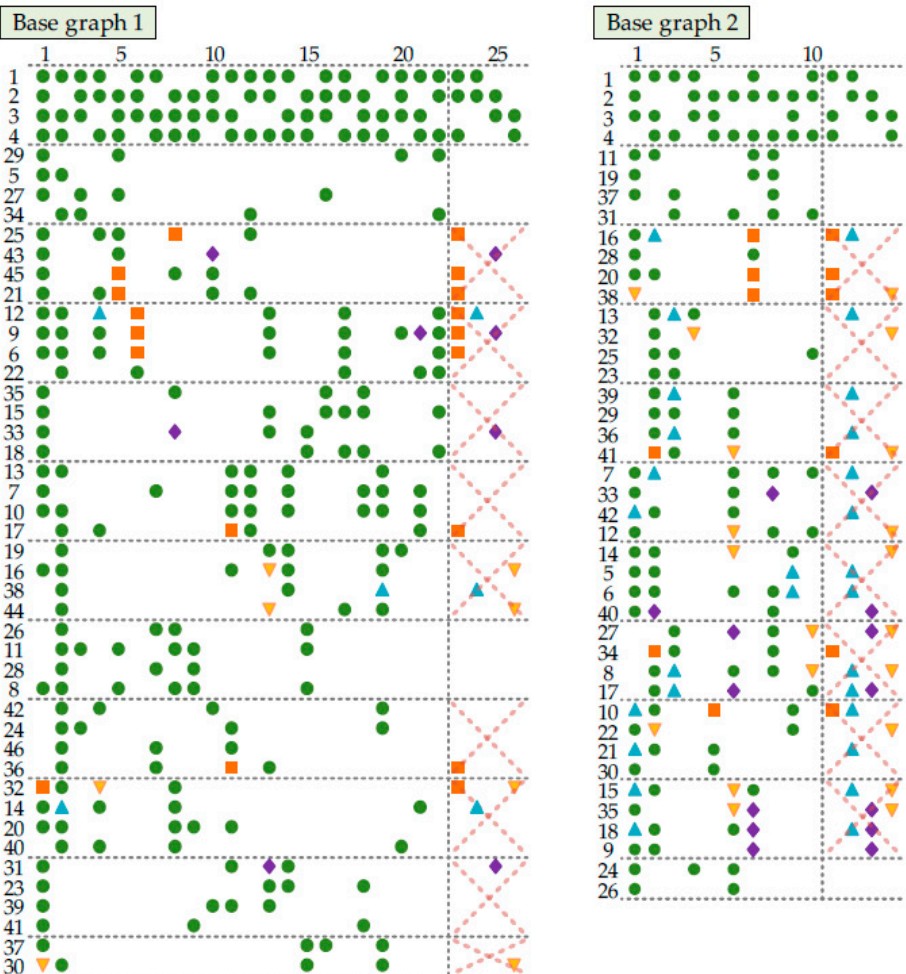

**Figure 12.** A scatter diagram of the optimized processing order for BG1 and BG2 codes when four CSs are available. Circles represent CPMs that are used for information bits processing, whereas other shapes represent CPMs that are used for core parity bits processing. All core parity bits processing is merged with some information bits processing when free CSs are available.

Table 2 summarizes the results of GA optimization. The improvement metric describes how much faster the optimized schedule is, compared to the unoptimized. A large reduction in the number of clock cycles necessary for one codeword encoding can be observed. The BG2 optimization provides better improvements because a larger percentage of total CPMs are located in the core parity columns and since all these CPMs are processed together with other CPMs from information bit columns.

**Table 2.** Results of the GA optimization.

| Encoding Schedule | $N_{CPC}$ | | $N_{CPC, Optimized}$ | | Improvement wrt Unoptimized (%) | | Improvement wrt Serial (%) | |
|---|---|---|---|---|---|---|---|---|
| | BG1 | BG2 | BG1 | BG2 | BG1 | BG2 | BG1 | BG2 |
| Serial | 265 | 150 | - | - | - | - | 0 | 0 |
| Serial (2 CSs) | 223 | 136 | 165 | 86 | 35.15 | 58.14 | 60.61 | 74.42 |
| Serial (4 CSs) | 180 | 94 | 107 | 53 | 68.22 | 77.35 | 147.66 | 183.02 |

Moreover, the improvement is calculated with respect to the original serial processing. The results show that the proposed architecture granted almost three times faster encoding for codes with lifting sizes of 96 and smaller, and almost twice as faster for codes with lifting sizes between 96 and 192. The hardware overhead is expected to be small and comes

mainly because of the shifting network's flexibility, which will be shown in Section 6.2. Therefore, it is expected that the HUE will be significantly increased.

### 6.2. Throughput-, Latency-, and Hardware-Usage Efficiency Results

The proposed encoder was implemented on the Xilinx ZCU111 development board with the Zynq UltraScale+ RF-SoC device (XCZU28DR). The obtained maximum clock frequency was $f_{max}$ = 580 MHz. Throughput results for both BG1 and BG2 codes (the entire PCM) are summarized in Figure 13a. Results are shown for all available lifting size values. The information throughput is calculated as in (32). The peak BG1 throughput is 18.51 Gb/s, whereas the peak throughput for BG2 is 14.87 Gb/s. Figure 13a shows a significant improvement of the throughput results for lifting sizes smaller than or equal to 192 when the proposed flexible partially parallel encoder is compared to the serial processing. High throughput values are achievable for much smaller lifting size values since CS resources are better used for parallelism increase. The benefits of processing order optimization are also easily noticeable.

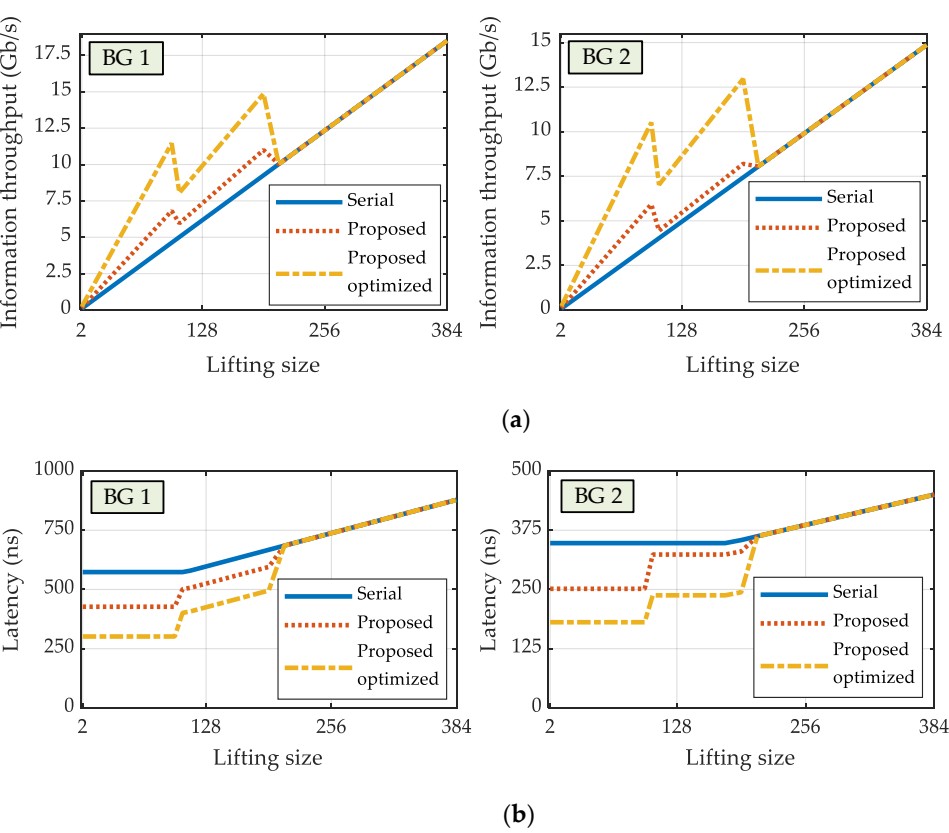

**Figure 13.** Performance of the proposed 5G NR LDPC encoder for three different encoding schedules and for various lifting sizes: (**a**) information throughput and (**b**) latency.

Encoder's latency was calculated as the time between the last transfer of the input information bits data and the last output codeword data transfer. The obtained latency is smaller than 1 µs for all codes. Results for all available lifting size values are shown in Figure 13b.

In order to compare obtained results with the state of the art, three additional 5G NR LDPC encoder cores were implemented as follows: (1) encoder that incorporates serial CPM processing whose architecture is presented in this paper but would be a conventional way to encode the 5G NR codes, (2) encoder that incorporates column-based parallel processing (a method described in [34]), and (3) encoder that incorporates row-based parallel processing (described in [31,35]). A detailed explanation of all the processing schedules used in the abovementioned architectures is given in Section 4.1.

The serial encoder has the same architecture as the proposed encoder. However, all redundant hardware components were removed. The CS was implemented as in [11], whereas blocks for merging and selection of λ values and core parity bits were removed. Additional simplification was made in the storage of PCM matrix data since there was no need for the storage of optimized schedules.

As described in Section 4.1, the column-based parallel architecture has been previously used for encoding Wi-Fi LDPC codes [34]. For the sake of comparison, the same architecture was applied to 5G NR LDPC codes. All columns in the exponent matrix are processed in parallel, which implies the usage of 46 CSs and GF(2) accumulation circuits. CSs were the same as for the serial architecture. Due to the pipeline, it was necessary to wait for two clock cycles for core parity bits to be ready, which increased the total number of clock cycles estimated in Section 4.1 to 28 for BG1 and 16 for BG2. This LDPC encoder core also calculates core parity bits using the same hardware as the proposed partially parallel architecture. PCM matrix was stored in the block RAM memories (BRAMs) in the form of eight exponent matrices for the highest lifting size, whereas the shift values for smaller lifting sizes were calculated in logic. It is necessary to load up to 46 shift values in parallel; hence multiple BRAMs were necessary to store these values.

To the best of the authors' knowledge, the only architecture applied to 5G NR LDPC encoding so far was row parallel architecture [35]. As described in Section 4.1, all rows of the exponent matrix are processed in parallel. Such an approach requires the usage of 26 CSs. In contrast to [35], all shifters in the present implementation were flexible (the same as in serial and column parallel architecture), i.e., they supported all lifting sizes from the standard. Shifters' outputs are summed in parallel using a tree-structured XOR summation. The PCM matrix was also stored in the block RAM memories in the same form as for column parallel architecture. However, since a smaller number of shift values is necessary in parallel, the number of used BRAMs is also smaller. Core parity bits were also calculated from λ vectors as in former designs.

Implementation results and comparison are given in Table 3. Given resources are without input and output buffers. Therefore, utilized BRAMs are only used for PCM storage. The obtained maximum operating frequency differed depending on the chosen architecture as shown in Table 3. Although they are heavily pipelined, larger designs have smaller maximal operating frequencies due to routing congestion. However, in order to make a better architectural comparison, the information throughput was normalized with the operating frequency as follows:

$$Thr_{norm} = \frac{Thr}{f_{CLK}}. \tag{35}$$

Normalized throughput values are calculated for both BG1 and BG2 and for all lifting sizes. The average normalized throughput values and the average throughputs are shown in Table 3. Finally, the HUE is calculated as throughput, both normalized and achieved, divided by the number of utilized hardware resources. Results show that, on average, the proposed architecture provides by far the highest HUE.

Finally, Figure 14 shows the normalized HUE depending on the lifting size for all the above architectures. It is obvious that highly parallel architectures are very inefficient, when compared to the serial and proposed partially parallel architecture. Considering the fact that the latency of the serial processing architecture and the proposed partially parallel architecture is lower than 1 μs for all codeword lengths, and the fact that the air interface requirement for 5G ultra-reliable low-latency communication (URLLC) use case is 1 ms [47], which is 1000 times higher, there is no remaining reason to use highly parallel architectures for 5G LDPC encoding. By placing multiple encoders to work in parallel, the proposed architecture can be used for obtaining the throughput that is even higher than the throughput of the highly parallel architectures but with much fewer hardware resources.

**Table 3.** Implementation results for various architectures of 5G NR LDPC encoder and average throughput and hardware usage efficiency comparison.

| Architecture | Resources Utilization | | | | $Thr_{norm,avg}$ (b/Cycle) | $f_{max}$ (MHz) | $Thr_{avg}$ (Gbps) | Avg. HUE ($Thr_{(norm),avg}$/Resources) | | | |
|---|---|---|---|---|---|---|---|---|---|---|---|
| | Slices | LUTs | FFs | 36k BRAMs | | | | b/Cycle/ kSlice | b/Cycle/ kLUT | b/Cycle/ kFF | Gbps/ kSlice |
| Column based parallel [34] | 42,257 | 248,641 | 246,960 | 6 | 61.9 | 330 | 20.43 | 1.46 | 0.25 | 0.25 | 0.48 |
| Row based parallel [31,35] | 26,180 | 126,986 | 131,185 | 3.5 | 30.1 | 470 | 14.15 | 1.15 | 0.24 | 0.23 | 0.54 |
| Serial | 1729 | 8712 | 9975 | 3 | 6.6 | 580 | 3.83 | 3.82 | 0.76 | 0.66 | 2.22 |
| Proposed optimized | 2215 | 11156 | 9635 | 5 | 10.3 | 580 | 5.97 | **4.65** | **0.92** | **1.07** | **2.70** |

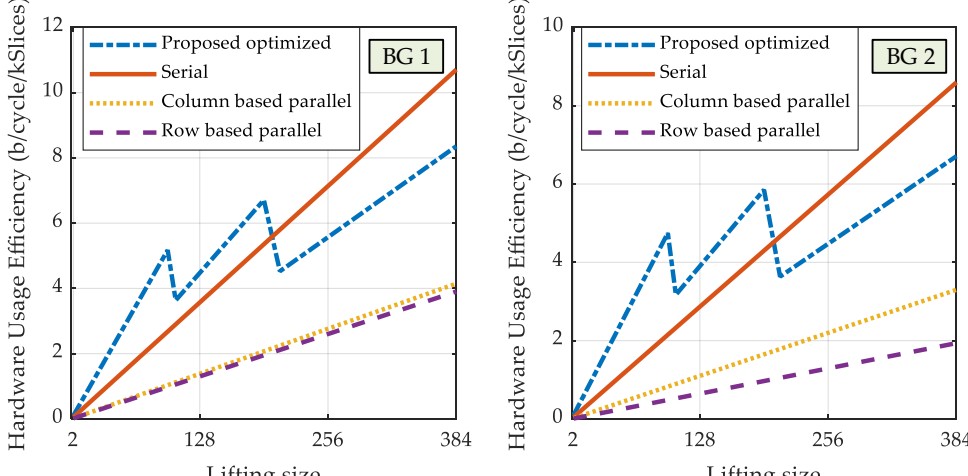

**Figure 14.** Normalized hardware usage efficiency for various 5G NR LDPC encoder architectures depending on the lifting size.

## 7. Conclusions

In this paper, a novel architecture for 5G NR LDPC encoding is presented. The designed encoder can be configured to encode all LDPC codes from the 5G NR standard in runtime. The architecture is based on the serial encoding schedule that processes one CPM of the PCM per clock cycle. However, it incorporates a novel approach of reusing available hardware resources for processing multiple CPMs per clock cycle for codes with shorter codewords. This is obtained by the innovative design of the circular shifting network which can be configured to work as a single shifter for long input data vectors or as multiple independent CSs for shorter inputs. Proposed CS architecture is also applicable in other hardware designs for both encoding and decoding of LDPC codes when flexibility is a requirement.

In addition to the architectural contributions, the paper also presented the method for offline encoding schedule optimization when multiple CPMs are processed per clock cycle. The GA-based optimization provided a great improvement in the encoding throughput (up to 183%) and latency (up to 92%). Implemented encoder provided multi-gigabit throughput and showed significant improvement in hardware usage efficiency, when compared with other architectures used for encoding of QC–LDPC codes. On average, the proposed encoder implementation has five times higher HUE, when compared to highly parallel implementations and about 22% higher HUE, when compared to the serial implementation. The obtained throughput and latency match the required specifications for all currently proposed 5G use cases. The flexibility of the proposed encoder makes it applicable in any of those scenarios.

Some contributions described in the paper can be applied in encoders or decoders for other QC–LDPC codes. The GA-based optimization can be performed for any architecture

that requires multiple rows processing for possible throughput increase. In addition to that, the proposed cyclic shifter design can be used in any encoder or decoder design that needs multiple shifting modes. A good example would be the encoder for WiMAX LDPC codes whose lifting sizes vary from 24 to 96.

**Author Contributions:** Conceptualization, V.L.P., D.M.E.M., and A.R.; methodology, V.L.P. and D.M.E.M.; validation, V.L.P.; formal analysis, V.L.P. and D.M.E.M.; investigation, V.L.P.; resources, A.R.; writing—original draft preparation, V.L.P., D.M.E.M., and A.R.; visualization, V.L.P.; software, V.L.P. and D.M.E.M.; supervision, A.R.; project administration, A.R.; funding acquisition, A.R. and V.L.P. All authors have read and agreed to the published version of the manuscript.

**Funding:** This research received no external funding.

**Acknowledgments:** Authors are thankful to Miloš Marković from the Tannera LLC and Lazar Saranovac, Srđan Brkić, Đorđe Sarač, and Predrag Ivaniš from the University of Belgrade, School of Electrical Engineering for valuable comments and discussions.

**Conflicts of Interest:** The authors declare no conflict of interest.

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
