# Peer review of "Flexible 5G New Radio LDPC Encoder Optimized for High Hardware Usage Efficiency"

_electronics, doi:10.3390/electronics10091106_

Round 1

Reviewer 1 Report

The manuscript presents an architecture for an LDPC encoder (5G NR) that its is efficient with respect to the hardware utilization. The paper is well written, organized and presented. It is overall kind of long but on the other hand it is explanatory and all concepts/techniques clear to the reader.

Author Response

We sincerely thank you for your review and comments. Regarding the length of the paper, we agree that it could have been shorter. However, we wanted to provide a thorough summary of the previous methods for encoding of LDPC codes, not only for the purpose of presenting our results, but for possible readers' general interest in this subject.

Reviewer 2 Report

The paper has been examined in detail. Overall, the work has been conducted on a well-investigated subject, which has extensive available literature. The authors should clearly elaborate on the specific aspects of their proposed framework that have been effective/efficient for the enabling technologies of 5G-centric applications (e.g., mmWaves, directional transmission, beamforming, massive MIMO, improved spectral efficiency, throughput, modulation, etc.); as given in the title/abstract but is missing from the contents; and should further highlight the key aspects that their proposed coding scheme based on LDPC in a more comprehensive qualitative way, both in Table 3 as well as in the Introduction part (background study). Also, the textbook-wise materials in Section 2 should be discussed in more detail and should reflect on its relevancy for better understanding the proposed coding scheme itself. In this regard, it should be described whether this coding scheme can be also applied to the current 4G/LTE systems and other potential wireless/telecom applications. The English language and technical presentation should also be reviewed and improved.

Author Response

We sincerely thank you for your review and all your comments and suggestions. Please, find our responses to your concerns attached as PDF file.

Reviewer 3 Report

Comments to the manuscript:
“Flexible 5G New Radio LDPC Encoder Optimized for High Hardware Usage Efficiency”

The manuscript “Flexible 5G New Radio LDPC Encoder Optimized for High Hardware Usage Efficiency” presents a novel solution for hardware implementation of the encoding method: partially parallel architecture that can provide high hardware usage of all 5G NR codes.

The presented idea is interesting and developmental, especially from a practical implementation point of view. Article content is up-to-date, noteworthy, and appropriate to Electronics. This version of the manuscript does not need corrections before publication. Below, in detail, I described my suggestions that may help to improve this article.

  1. Title

The title should reflect its main idea, e.g., a specific approach, method, scenario, its novelty aspect, etc. Generally, the title of the reviewed paper reflects well the paper contribution.

  1. Abbreviation

Generally, the authors explain most of the used abbreviations, but there are some missing or comments:

  • Page 4; line 139 :abbreviation in the subtitle or full method name RU was introduced in the "Introduction"-not such need to secondly describing the abbreviation.
  • Page 4; line 154: abbreviation for Galois field - GF is missing.
  • Page 14; line 470: abbreviation for QSN is missing.

  1. Content

The introduction provides sufficient background and includes relevant references. The research design is appropriate, and the method is adequately described. The conclusions are supported by the results, which are clearly presented.

English language and style are acceptable, but maybe the authors might correct:

  • Page 2; line 51: “code structure can significantly affect the HUE” - consider adding an article.
  • Page 10; line 347: “However, the required number of…”- consider adding an article.
  • Page 10; line 326: “Encoding throughput can be improved by increasing the parallelism.” – consider the change.
  • Page 11; line 365-367: Sentence may be hard to follow. Consider rephrasing: “This happens because many shifters are unused for many rows in rows based parallel architectures and for many columns in column-based parallel architectures.”
  • Page 16; line 522-523:”…other LDPC encoder or decoder architecture could possibly have more benefits…” – could possibly may be redundant - consider the change.
  •  

Others:

  • Page 2; line 90: “..which highly increases hardware usage efficiency.” – and might affect power consumption/battery life – next advantage of the Authors solution.
  • Page 2; line 93: ” Encoding schedules are optimized for minimum encoding time… ” - do Authors consider other criteria, for example, minimum/acceptable latency of the system?
  • Page 14; line 479: “where D is an integer number” – probably D is a natural number. If it is true, please correct it instead of “integer”?
  • Page 16; line 522-523:”…other LDPC encoder or decoder architecture could possibly have more benefits…” - could you explain those benefits.

  1. References

Reference [1]:”3rd Generation Partnership Project; Technical Specification Group Radio Access Network; NR; Multiplexing and channel coding (Release 16), 3GPP TS 38.212 V16.1.0 (2020-03), 2020.” - it is not actual. The actual is V16.5.0 published 30.03.2021, and it is good to refer to the most actual version.  In fact, from V16.1.0 there are four new releases that do not affect the actuality of the article.

Author Response

We sincerely thank you for your review and all your comments and suggestions. Please, find our responses to your suggestions attached as PDF file.

Round 2

Reviewer 2 Report

The revised version of this article has been examined in detail. The major points raised in the former review have been addressed in this version, and the current text may proceed with the publication. A final English check should be carried out to ensure the text is error-free in terms of grammar and presentation.